# Assessing Patient Understanding and Adherence to Preoperative Medication Advice Provided in Pre-Admission Clinic

**DOI:** 10.3390/healthcare13192429

**Published:** 2025-09-25

**Authors:** Alison Tse, Yasmin Baghdadi, Phan Tuong Van Nguyen, Rand Sarhan, Vivek B. Nooney, Wejdan Shahin, Andrew Vuong

**Affiliations:** 1Pharmacy Department, Western Health, St Albans, VIC 3021, Australia; yasmin.baghdadi@wh.org.au (Y.B.); andrew.vuong@wh.org.au (A.V.); 2Discipline of Pharmacy, RMIT University, Melbourne, VIC 3083, Australia; vantpnguyen@gmail.com (P.T.V.N.); randsarhan94@gmail.com (R.S.); vivek.nooney@rmit.edu.au (V.B.N.); wejdan.shahin@rmit.edu.au (W.S.)

**Keywords:** preoperative care, medication advice, medication adherence, communication

## Abstract

**Background:** Appropriate medication management before surgery is essential to minimise perioperative risk. Patient adherence to preoperative medication advice demonstrates considerable variability and is influenced by multiple interacting factors. This study assessed patient understanding and adherence to preoperative medication advice provided in the Pre-Admission Clinic (PAC) and identified factors contributing to non-adherence. **Methods:** A cross-sectional survey study was conducted over 12 weeks in 2022 at a tertiary hospital. Adult patients scheduled for elective surgery who received preoperative medication advice in PAC were surveyed on the day of surgery. Data collected included demographics, clinical characteristics, adherence, reasons for non-adherence, and communication preferences. Descriptive and inferential statistics were used for analysis. **Results:** Of 156 participants, 91 (58.3%) adhered to medication advice, while 65 (41.7%) did not. Common reasons for non-adherence included forgotten advice (35.4%), misunderstood advice (33.8%), and intentional deviation due to surgery (18.5%). Non-adherence rates were highest for NSAIDs (50.0%) and P2Y12 inhibitors (45.5%). Two surgeries were cancelled due to the delayed cessation of anticoagulants. Non-adherence was significantly associated with a greater number of medications requiring perioperative management (*p* = 0.004) and a longer duration between PAC and surgery (*p* = 0.010). Most non-adherent patients (64.7%) preferred a combination of verbal and written advice. **Conclusions:** A substantial proportion of patients were non-adherent to preoperative medication advice, often due to unclear communication or a lack of understanding of the clinical rationale for the advice. Multimodal strategies, including written or digital reinforcement of verbal advice, multidisciplinary collaboration, and patient-centred education, may improve adherence and reduce preventable cancellations. Future studies should evaluate the impact of these interventions.

## 1. Introduction

Preparation for successful surgery requires a comprehensive multidisciplinary assessment and planning. For elective surgeries, preoperative assessment is conducted in the Pre-Admission Clinic (PAC) to ensure patients are optimally prepared. Medication reconciliation and the provision of preoperative medication advice are essential to minimising risks associated with surgery and anaesthesia. Non-adherence to this advice may contribute to perioperative complications or surgical cancellation, with potential clinical, operational, and financial impacts [1,2,3,4,5].

Failure to appropriately withhold antiplatelets or anticoagulants prior to surgery has been associated with an increased risk of perioperative bleeding. The use of direct oral anticoagulants has been linked to a haemorrhage rate of 10.8% in surgical patients, compared to an overall rate of 0.8% [6]. Conversely, stopping anticoagulants too early or unnecessarily in patients with atrial fibrillation has been shown to elevate the risk of thromboembolic events from 1.3% to 15.2% [6]. Anti-hyperglycaemic agents also pose significant perioperative risks if not managed appropriately. Multiple studies have demonstrated an increased risk of euglycaemic diabetic ketoacidosis associated with sodium-glucose co-transporter-2 (SGLT2) inhibitors, particularly in the context of fasting or surgical stress [7,8,9]. In the study by Iwasaki et al. [7], 22.6% of surgical patients treated with SGLT2 inhibitors developed euglycaemic diabetic ketoacidosis postoperatively, compared to 8.1% of those not treated with SGLT2 inhibitors. Additionally, disease-modifying antirheumatic drugs (DMARDs) can increase the risk of postoperative surgical site infections and delayed wound healing due to their immunosuppressive and immunomodulatory effects [10,11]. These findings underscore the importance of appropriate perioperative medication management to minimise the risk of adverse outcomes in surgical patients.

At our institution, medication reconciliation in the PAC may be performed by a nurse, pharmacist, surgical doctor, and/or anaesthetist. Preoperative medication advice is typically provided by a nurse, a surgical doctor, and/or an anaesthetist. Following the onset of the COVID-19 pandemic, the majority of PAC consultations transitioned to telehealth, with preoperative medication advice most commonly delivered verbally, either in person at PAC or via telephone. Written instructions (e.g., SMS, email, or printed handouts) are not routinely provided, although patients may occasionally receive them. This reliance on verbal advice may contribute to variability in patient recall and adherence. Observations at our institution indicate that poor adherence to this advice has been a persistent issue, occasionally leading to surgical cancellations. Understanding the contributors to non-adherence is fundamental to informing the development and implementation of targeted improvement strategies. While some literature highlights medication non-adherence or mismanagement before surgery, there are limited studies that evaluate the risk factors and underlying causes of non-adherence to medication advice involving specific preoperative adjustments [12,13,14].

The aim of this study was to assess patients’ understanding of, and adherence to, preoperative medication advice provided in PAC. The primary objectives were to quantify the proportion of patients who did not adhere to the advice and evaluate the contributing factors. A secondary objective was to explore patients’ preferred methods of communication for receiving preoperative medication advice.

## 2. Materials and Methods

### 2.1. Study Design

We conducted a single-centre cross-sectional survey study at a tertiary institution in Melbourne, Australia. The study was undertaken over a 12-week period from July to October 2022. This Quality Assurance Research was conducted after obtaining approval from the Western Health Low Risk Ethics Panel. The organisational approval was granted on 14 July 2022 (Reference number: QA/86539/WH-2022-320350(v2)). Verbal informed consent was obtained from all participants prior to their participation in the study.

### 2.2. Participants

A convenience sampling method was used to recruit 150 to 200 patients. This approach was appropriate for a quality assurance project where the primary aim was to capture real-world adherence behaviours in the perioperative setting. The inclusion criteria included adult (18 years and above) patients scheduled for elective surgery who attended PAC and received medication advice for one or more of the following classes based on our institution’s guidelines for medication requiring perioperative management:Antiplatelets or anticoagulants;Non-steroidal anti-inflammatory drugs (NSAIDs);Anti-hyperglycaemic agents;Disease-modifying antirheumatic drugs (DMARDs).

Patients with cognitive impairment who did not manage their own medications were included in the study, with consent obtained from their carer, power of attorney (POA), or next of kin (NOK) responsible for medication management. Patients who declined to participate and those with cognitive impairment who managed their own medications were excluded.

### 2.3. Procedures

Theatre lists were screened by pharmacists and pharmacy student associate investigators between 1 and 3 days prior to the day of surgery. Patient Electronic Medical Records (EMRs) were reviewed to determine if patients had attended a PAC and received advice for one or more medicines requiring preoperative management. Eligible patients were approached by investigators on the day of surgery. Verbal consent was obtained from patients, or from their carer, power of attorney (POA), or next of kin (NOK), where applicable, in a private consultation area. Patients with cognitive impairment were excluded if their carer, POA, or NOK was not present on the day or could not be contacted. Interpreter services, either in-house or via phone, were used to assist with communication for patients from non-English-speaking backgrounds.

The survey comprised 6 structured questions, including both closed- and open-ended formats (Appendix A). The questions assessed included patient recall of the preoperative medication advice provided in the PAC; whether the patient perceived the advice to be clear; actual medication management in the preoperative period regardless of the advice recalled by the patient; reasons for non-adherence; and the preferred method of communication.

Adherence was defined as full compliance with all preoperative medication instructions provided in PAC. Patients who failed to follow one or more instructions were classified as non-adherent. A binary approach was applied even when multiple medications were involved, as partial adherence still presented a risk for surgical cancellation. When non-adherence to preoperative medication advice was identified, further data were collected to classify the cause of non-adherence, including medication stopped on the day of surgery; medication stopped too early; medication stopped too late; and medication not stopped. To establish content validity, the questionnaire was piloted with five clinical pharmacists of varying experience at our institution. Two pharmacists had more than 10 years’ experience, one pharmacist had between 5 and 10 years’ experience, and two pharmacists had fewer than 5 years’ experience. Two pharmacists had experience in perioperative medicine. All pharmacists agreed that the items were clear, relevant, and feasible for administration in the perioperative setting, and no changes were required following pilot testing.

### 2.4. Variables

Demographic and clinical data, including age, gender, primary language, primary medication manager, use of a dose administration aid, surgical unit, advice provider, duration between PAC appointment and surgery, number of usual medications, and the number and type of medications requiring preoperative management, were extracted from the EMR.

### 2.5. Data Collection

Prospective data were collected via participant survey on the day of surgery. Retrospective data, including demographic and clinical variables, were collected from EMR. Prospective and retrospective data were collected by the research project team, comprising pharmacy student associate investigators and pharmacists employed at the institution. Data were recorded electronically using Research Electronic Data Capture (REDCap), a web-based application designed to facilitate data collection and management whilst maintaining strong data protection [15]. Data validation strategies were utilised in the development of the data collection tool to reduce the risk of missing or erroneous data entry.

### 2.6. Primary Aims

The primary aims were to determine the proportion of patients who were non-adherent to preoperative medication advice provided in PAC and to evaluate the contributing factors to non-adherence.

### 2.7. Secondary Aims

The secondary aim was to determine patients’ preferred methods of communication for receiving preoperative medication advice.

### 2.8. Statistical Analysis

Data entered in REDCap were exported into IBM SPSS Statistics (version 28) for analysis. Pivot tables in Microsoft Excel were also used for preliminary descriptive summaries.

Survey responses were analysed using descriptive statistics. Demographic and clinical characteristics were summarised and compared between adherent and non-adherent patient groups. Categorical variables were compared using the Chi-square test, and continuous variables using the independent sample *t*-test. A *p*-value ≤ 0.05 was considered statistically significant.

## 3. Results

During the study period, 158 patients who met the inclusion criteria were approached by investigators, and 156 (98.7%) consented to participate in the study. Patient demographic and clinical characteristics are presented in Table 1.

Of the 156 patients who consented to participate in the study, 104 (66.7%) correctly recalled the preoperative medication advice provided, while 52 (33.3%) either recalled it incorrectly or were unable to recall it. Among all 156 patients, 91 (58.3%) were adherent to the preoperative medication advice, whereas 65 (41.7%) were non-adherent. Of the 65 non-adherent patients, the most common reasons were: forgotten advice (35.4%), misunderstood advice (33.8%), and intentional deviation related to surgery (18.5%). Additional reasons included receiving conflicting advice from different providers (7.7%) and general non-adherence to their usual medications (4.6%) (Figure 1). Non-adherent patients managed more medications requiring preoperative adjustment (mean 2.20 vs. 1.74; *p* = 0.004) and experienced a longer duration between their PAC appointment and surgery (15 vs. 10 days; *p* = 0.010), compared to adherent patients.

Adherence rates and deviations from preoperative medication advice varied across medication types. Non-adherence was most prevalent for NSAIDs (50.0%) and P2Y12 inhibitors (45.5%), followed by SGLT2 inhibitors (36.8%) and aspirin (34.4%). Among all non-adherent cases, 49 (60.5%) involved medications being stopped too early, while 27 (33.3%) involved medications being stopped too late. The remaining 5 cases (6.2%) involved incorrect aspirin bridging in patients who were normally treated with a P2Y12 inhibitor, or the administration of an incorrect insulin dose. Delayed cessation of antiplatelet or anticoagulant therapy was observed in 10 cases, resulting in two cancellations on the day of surgery. Conversely, aspirin and anticoagulants were frequently stopped too early or unnecessarily, accounting for 18 of 21 cases (85.7%) and 9 of 13 cases (69.2%) of non-adherence, respectively (Table 2).

Among all 156 patients, 137 (87.8%) received verbal advice only, with the remaining 19 (12.2%) receiving both verbal and written preoperative medication instructions. This was in contrast to participant preference for a combination of both verbal and written advice (60.9%), followed by verbal advice only (19.9%) and written advice only (18.6%). Communication preferences were not specified for one patient (0.6%). Among the non-adherent group (n = 65), combined verbal and written communication was preferred (64.7%), followed by written only (20.0%) and verbal only (13.8%) advice (Figure 2). The proportion of participants preferring combined written and verbal advice was comparable between those whose primary language was English and those whose primary language was not English (60% vs. 67%, *p* = 0.56).

## 4. Discussion

This study identified a high rate of non-adherence to preoperative medication advice provided in PAC, with 41.7% of patients failing to follow the advice. The rate of non-adherence is similar to that documented in some studies in the literature [12,13]. Paul et al. (2022) reported a non-compliance rate of 37.2% among Canadian adults undergoing elective surgery, highlighting a comparable burden of preoperative medication non-adherence across healthcare systems [12]. Carroll et al. (2007) showed that patient recall of perioperative instructions decays rapidly when advice is provided verbally only [16]. In this study, the primary reasons for non-adherence were related to cognitive factors, specifically forgotten advice (35.4%) and misunderstood advice (33.8%). These findings indicate that the advice communicated to patients may have been unclear or inadequately conveyed, emphasising the inherent limitations of verbal communication alone, particularly when complex information is delivered by phone. This provides measurable evidence of a systemic issue across institutions, reinforcing the need for targeted interventions and quality improvement strategies.

A meta-analysis demonstrated that poor communication was associated with a 19% increased risk of non-adherence to medication advice compared to clear and concise communication [17]. Our data showed that more than 84% of non-adherent patients preferred receiving written information, either independently or combined with verbal advice, with 64.7% specifically favouring the combination of both verbal and written formats. This strongly supports the implementation of multimodal communication strategies in PAC workflows. Several studies have demonstrated that the use of clear and standardised written advice can improve adherence, particularly with short-term medication changes in the preoperative setting [18,19]. These results are supported by Selic et al. (2011), who showed that patients receiving both verbal and written instructions recalled advice more accurately and reported higher satisfaction [20]. Ong et al. (2023) also found that structured written reinforcement improved compliance from 38% to 71% [21]. Reinforcing verbal instructions with written material could be a practical and effective strategy to improve understanding and retention.

Variability in adherence was observed across different medication types. The most frequent deviation was stopping medications too early, particularly with aspirin and anticoagulants. Many of these patients correctly recalled the advice but chose to stop the medication unnecessarily or earlier than advised, often influenced by prior surgical experience or concerns about bleeding. Siddique et al. (2023) similarly reported that patients on anticoagulants deviated from preoperative recommendations, often by stopping therapy too early [22]. This aligns with the finding that intentional deviation accounted for nearly one-fifth of non-adherent patients (18.5%), suggesting that some patients lacked confidence in the clinical rationale communicated by the advice provider or prioritized their personal judgment regarding surgical needs. Conversely, 10 cases of non-adherence were related to stopping antiplatelets or anticoagulants too late, resulting in two cancellations on the day of surgery. These findings indicate gaps in patient education related to the risks associated with their surgery and the rationale behind the recommended preoperative medication management. Providing patient-centred and appropriate education may support improved adherence.

No statistically significant demographic differences were observed between adherent and non-adherent groups, suggesting that factors such as age, gender, primary language, or medication management approach are unlikely to have an impact on adherence. However, non-adherent patients were managing more medications requiring preoperative adjustment (mean 2.20 vs. 1.74; *p* = 0.004) and experienced a longer duration between their PAC appointment and surgery (15 vs. 10 days; *p* = 0.010). These findings indicate that medication complexity and time delays may increase the risk of non-adherence, and multimodal communication strategies should be utilised. Additionally, although the type of advice provider (nurse, surgical doctor, anaesthetist) did not influence adherence, variability in the communicated advice from multiple providers highlights deficiencies in multidisciplinary coordination that may contribute to patient confusion, emphasising the need for standardised communication.

This study has several limitations. Data were collected from a single institution, which may limit the generalisation of the findings. The relatively small sample size may have also reduced the power to detect significant differences within certain demographic subgroups. Furthermore, data collected from survey responses may be subject to recall or response bias. The analysis assumed that the preoperative medication advice provided was consistent with documentation in the EMR. Clinical outcomes, such as perioperative complications related to non-adherence, were also not evaluated in this study. Since this study was conducted, preadmission practices have evolved. Pharmacists are now more frequently embedded in PAC, and the provision of written advice to supplement verbal medication instructions has become more common. These changes may improve consistency of communication and patient adherence, but they also mean our findings should be interpreted in the context of an evolving model of care.

To improve adherence to preoperative medication advice, several strategies should be considered. Providing written information during PAC consultations, such as EMR-generated or SMS-based medication advice, may reinforce verbal communication and reduce misunderstanding. Tailored patient education that explains the clinical rationale for medication advice may be particularly beneficial for individuals treated with blood-thinning agents or those with limited health literacy. A multidisciplinary approach involving nurses, pharmacists, surgical doctors, and anaesthetists is recommended to promote consistency in the information communicated and enhance patient comprehension. Greater standardisation of medication advice and compliance with established protocols are also needed and may be facilitated by the involvement of perioperative pharmacists within the PAC setting. Future research should evaluate the effectiveness of these interventions in improving adherence and minimising the associated cancellation of surgeries.

## 5. Conclusions

The provision of medication advice aims to minimise clinical risks during the perioperative period. Non-adherence to preoperative medication advice provided in PAC may be intentional or unintentional and, in some cases, can result in cancellation of surgeries. The findings offer valuable insights into the preventable causes of non-adherence and highlight the importance of providing standardised written information together with education regarding the rationale for medication management changes during the perioperative period. Addressing these issues requires targeted improvements in communication strategies and the implementation of coordinated, multidisciplinary approaches to ensure patients receive clear, consistent, and comprehensible advice.

## Figures and Tables

**Figure 1 healthcare-13-02429-f001:**
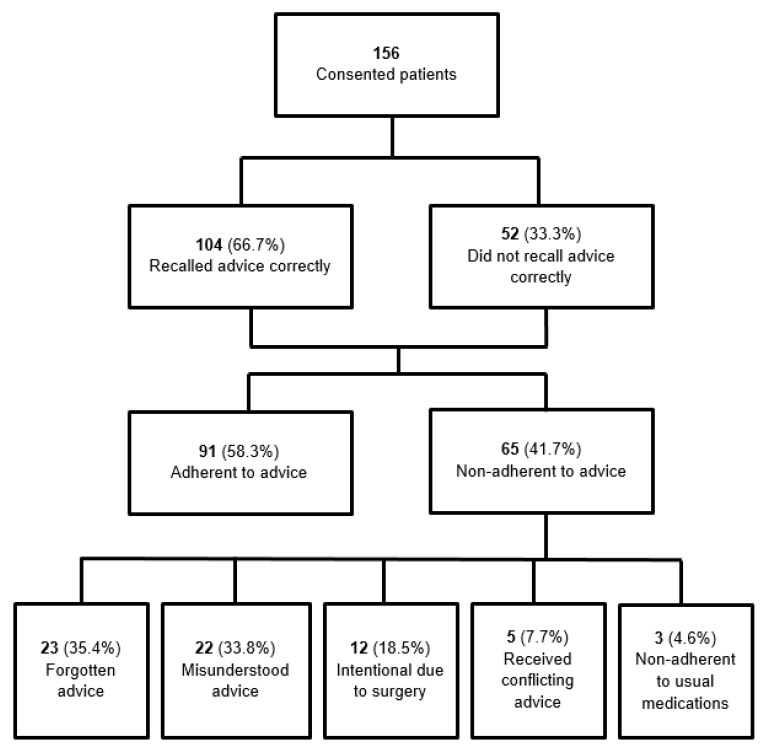
Summary of patient adherence to preoperative medication advice.

**Figure 2 healthcare-13-02429-f002:**
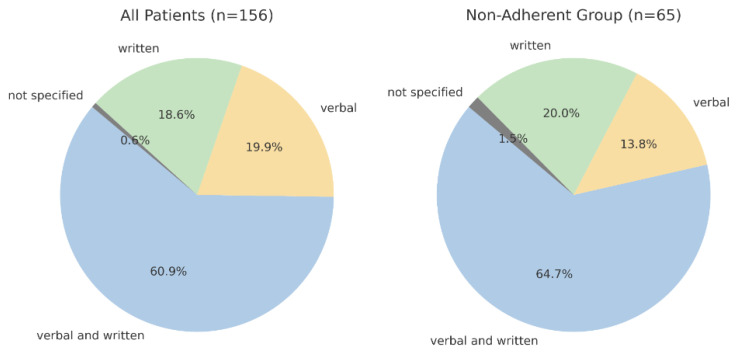
Communication preferences for preoperative medication advice.

**Table 1 healthcare-13-02429-t001:** Patient demographic and clinical characteristics.

Variable	Adherent Group (n = 91)	Non-Adherent Group (n = 65)	*p*-Value
Age (years)—Mean ± SD	67.7 ± 14.2	69.7 ± 12.0	0.338
Gender			0.642
Female	41 (45.1%)	26 (40.0%)	
Male	50 (54.9%	39 (60.0%)	
Primary language			0.552
English	77 (84.6%)	58 (89.2%)	
Non-English	14 (15.4%)	7 (10.8%)	
Primary medication manager			0.068
Patient self	76 (83.5%)	61 (93.8%)	
Carer	9 (9.9%)	4 (6.2%)	
Residential aged care	6 (6.6%)	0 (0.0%)	
Use of a dose administration aid			0.302
Used	26 (28.6%)	13 (20.0%)	
Not used	65 (71.4%)	52 (80.0%)	
Surgical Unit			0.243
General and Endocrine	6 (6.6%)	5 (7.7%)	
Otolaryngology	2 (2.2%)	0 (0.0%)	
Gynaecology	10 (11.0%)	3 (4.6%)	
Ophthalmology	6 (6.6%)	3 (4.6%)	
Orthopaedic	5 (5.5%)	9 (13.8%)	
Plastic	13 (14.3%)	7 (10.8%)	
Thoracic	2 (2.2%)	0 (0.0%)	
Upper Gastrointestinal/Hepatobiliary	3 (3.3%)	1 (1.5%)	
Gastroenterology	7 (7.7%)	11 (16.9%)	
Colorectal	13 (14.3%)	5 (7.7%)	
Urology	19 (20.9%)	18 (27.7%)	
Vascular	5 (5.5%)	3 (4.6%)	
Advice Provider *			0.792
Nurse	49 (58.3%)	35 (41.7%)	
Surgical doctor	43 (62.3%)	26 (37.7%)	
Anaesthetist	45 (57.0%)	34 (43.0%)	
Duration (days) between PAC appointment and surgery—Median (IQR)	10 (7–24)	15 (9–79)	0.010
Total number of usual medications—Mean ± SD, Median (IQR)	9.33 ± 4.77 8 (6–12)	10.22 ± 4.27 10 (8–12)	0.084
Number of usual medications requiring preoperative management—Mean ± SD, Median (IQR)	1.74 ± 1.13 1 (1–2)	2.20 ± 1.21 2 (1–3)	0.004

* Some patients received preoperative medication advice from multiple providers.

**Table 2 healthcare-13-02429-t002:** Adherence rates by medication types.

Medication Type	Adherent (n, %)	Non-Adherent (n, %)	Stopped Too Early (n)	Stopped Too Late (n)	Incorrect Dose Administered (n)	Incorrect Aspirin Bridging (n)
Aspirin	40 (65.6%)	21 (34.4%)	18	3		
P2Y12 inhibitors	12 (54.5%)	10 (45.5%)	4	3		3
Anticoagulants	32 (71.1%)	13 (28.9%)	9	4 *		
NSAIDs	9 (50.0%)	9 (50.0%)	2	7		
Oral hypoglycaemics (excluding SGLT2 inhibitors)	46 (74.2%)	16 (25.8%)	10	6		
SGLT2 inhibitors	12 (63.2%)	7 (36.8%)	4	3		
GLP-1/GIP receptor agonists	2 (100.0%)	0 (0.0%)				
Insulin	13 (76.5%)	4 (23.5%)	1	1	2	
DMARDs	3 (75.0%)	1 (25.0%)	1			

***** Resulted in two cancellations on the day of surgery.

## Data Availability

The data presented in this study are available on request from the corresponding author due to privacy, legal and ethical reasons.

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
