# Peer review of "Assessing Patient Understanding and Adherence to Preoperative Medication Advice Provided in Pre-Admission Clinic"

_healthcare, 2025, doi:10.3390/healthcare13192429_

Round 1

Reviewer 1 Report

Comments and Suggestions for Authors

The paper is a well-conceived and timely study addressing an important area in perioperative safety: patient adherence to preoperative medication advice. The authors highlight a relevant problem that has both clinical and operational consequences, namely, preventable surgical cancellations and perioperative complications due to inappropriate medication management. The study design (cross-sectional survey) is pragmatic, and the findings (41.7% non-adherence, associated with longer intervals from consultation to surgery and complexity of medication regimens) are clinically meaningful.

Nevertheless, there are areas that could be improved through revision.

1. The methodology section does not provide an adequate description of the survey instrument. The questionnaire is only briefly mentioned, without details on its structure, number or type of questions, or the rationale for their inclusion. This lack of transparency limits the reproducibility of the study and prevents readers from assessing the appropriateness of the tool.

Moreover, there is no reference to any validation of the questionnaire, either through pilot testing or by comparison with established instruments. The absence of information on reliability or validity raises concerns regarding the robustness of the collected data and the strength of the conclusions drawn.

2. An important methodological limitation is the lack of clarity regarding how patients were classified as “adherent” or “non-adherent.” The manuscript does not describe whether adherence was treated as a binary outcome only, or if partial adherence was considered. For instance, some patients may have followed certain instructions while failing to comply with others. Without explicit criteria, it is difficult to assess the accuracy of the reported adherence rates and the validity of the subsequent statistical comparisons.

Related to this, it remains unclear how patients taking multiple medications requiring perioperative management were handled in the analysis. If a patient adhered to recommendations for some medications but not others, how were they categorized? Were they classified as entirely non-adherent, or was a graded approach used? Clarifying these methodological choices is essential, as they may significantly affect the interpretation of adherence patterns across medication classes and the study’s overall conclusions.

Minor remark: There is a typographical error in line 54: ‘Additionally. disease-modifying antirheumatic…’.I hope these comments will be useful to the authors in revising and strengthening their manuscript.

Author Response

Comment 1: 

The paper is a well-conceived and timely study addressing an important area in perioperative safety: patient adherence to preoperative medication advice. The authors highlight a relevant problem that has both clinical and operational consequences, namely, preventable surgical cancellations and perioperative complications due to inappropriate medication management. The study design (cross-sectional survey) is pragmatic, and the findings (41.7% non-adherence, associated with longer intervals from consultation to surgery and complexity of medication regimens) are clinically meaningful.

Author response: Thank you for your comments. We addressed them all and believe the paper is substantially improved now.

Comment 2: 

Nevertheless, there are areas that could be improved through revision.

The methodology section does not provide an adequate description of the survey instrument. The questionnaire is only briefly mentioned, without details on its structure, number or type of questions, or the rationale for their inclusion. This lack of transparency limits the reproducibility of the study and prevents readers from assessing the appropriateness of the tool.

Author response: the paragraph below is added to the method section: “ The survey comprised 6 structured questions, including closed and open-ended formats. The questions assessed: patient recall of the preoperative medication advice; whether the patient perceived the advice to be clear; actual medication management in the preoperative period regardless of the advice recalled by the patient; reasons for non-adherence; whether verbal or written communication was preferred, or a combination of both; and whether the patient had any other comments about their experience with preoperative medication management. ”

Comment 3: Moreover, there is no reference to any validation of the questionnaire, either through pilot testing or by comparison with established instruments. The absence of information on reliability or validity raises concerns regarding the robustness of the collected data and the strength of the conclusions drawn.

Author response: this paragraph has been added to the methodology to address this comment: “To establish content validity, the questionnaire was piloted with five clinical pharmacists of varying experience at our institution. All pharmacists agreed that the items were clear, relevant, and feasible for administration in the perioperative setting, and no changes were required following pilot testing.”

Comment 4: An important methodological limitation is the lack of clarity regarding how patients were classified as “adherent” or “non-adherent.” The manuscript does not describe whether adherence was treated as a binary outcome only, or if partial adherence was considered. For instance, some patients may have followed certain instructions while failing to comply with others. Without explicit criteria, it is difficult to assess the accuracy of the reported adherence rates and the validity of the subsequent statistical comparisons.

Author response: This has been included in the method to clarify it:Adherence was defined as full compliance with all preoperative medication instructions. Patients who failed to follow at least one instruction were classified as non-adherent. A binary approach was applied even when multiple medications were involved, as partial adherence still presented a risk for surgical cancellation.”

Comment 5: Related to this, it remains unclear how patients taking multiple medications requiring perioperative management were handled in the analysis. If a patient adhered to recommendations for some medications but not others, how were they categorized? Were they classified as entirely non-adherent, or was a graded approach used? Clarifying these methodological choices is essential, as they may significantly affect the interpretation of adherence patterns across medication classes and the study’s overall conclusions.

Author response: This has been addressed as mentioned above and this has been included in the method section: “A binary approach was applied even when multiple medications were involved, as partial adherence still presented a risk for surgical cancellation”

Comment 6: Minor remark: There is a typographical error in line 54: ‘Additionally. disease-modifying antirheumatic…’.I hope these comments will be useful to the authors in revising and strengthening their manuscript.

Author response: This has been corrected. Thank you for your comments.

Reviewer 2 Report

Comments and Suggestions for Authors

Dear authors, it was pleasure reading your article on medication adherence in preoperative period. I believe that the results of your study are very good and highly significant for the daily practice of clinical pharmacists in hospitals. However, I do have a few suggestions that I hope you will consider: 

  1. Please consider including Pubmed Mesh terms in your key words in order to enable higher visibility of the article. For instance, instead of preoperative, you could write preoperative care, etc.
  2. Patients usually receive advice on preoperative medication only through conversation? They are not given any written document? Please make it more clear in the introduction.
  3. Can you explain why you aimed for 150-200 patients? Did you perform study power calculation?
  4. Which statistical programme did you use? Please provide the information in methodology section.
  5. I would like to see more studies that you compare your results with in the discussion section. Each result of the study should be compared with the results of other authors.

Author Response

Dear authors, it was pleasure reading your article on medication adherence in preoperative period. I believe that the results of your study are very good and highly significant for the daily practice of clinical pharmacists in hospitals. However, I do have a few suggestions that I hope you will consider: 

Thank you for your comments. We have addressed them all.

Comment 1: Please consider including Pubmed Mesh terms in your key words in order to enable higher visibility of the article. For instance, instead of preoperative, you could write preoperative care, etc.

Author response: This has been done as follows: Preoperative care; Medication advice; Medication adherence; Communication

Comment 2: Patients usually receive advice on preoperative medication only through conversation? They are not given any written document? Please make it more clear in the introduction.

Author response: this has been included in the introduction to make it clearer: “…..with preoperative medication advice most commonly delivered verbally, either in person at the Pre-Admission Clinic or via telephone. Written instructions (e.g., SMS, email, or printed handouts) are not routinely provided, although patients may occasionally receive them. This reliance on verbal advice may contribute to variability in patient recall and adherence.”

Comment 3: Can you explain why you aimed for 150-200 patients? Did you perform study power calculation?

Author response: this has been included in the method section to make it clearer: “A convenience sampling method was used to recruit 150 to 200 patients. This approach was appropriate for a quality assurance project where the primary aim was to capture real-world adherence behaviours in the perioperative setting.”

Comment 4: Which statistical programme did you use? Please provide the information in methodology section.

Author response: this has been included in the method section: “Data were initially collated in REDCap and exported into IBM SPSS Statistics (version 28) for analysis. Pivot tables in Microsoft Excel were also used for preliminary descriptive summaries.”

Comment 5: I would like to see more studies that you compare your results with in the discussion section. Each result of the study should be compared with the results of other authors.

Author response: We have incorporated four additional references to strengthen the comparison of our results with existing literature. Please refer to the revised Discussion section for all the edits made.